# Center of Pressure Behavior in Response to Unexpected Base of Support Shifting: A New Objective Tool for Dynamic Balance Assessment

**DOI:** 10.3390/s23136203

**Published:** 2023-07-06

**Authors:** Alex Rizzato, Matteo Bozzato, Giuseppe Zullo, Antonio Paoli, Giuseppe Marcolin

**Affiliations:** 1Department of Biomedical Sciences, University of Padova, 35131 Padua, Italy; alex.rizzato@unipd.it (A.R.); matteo.bozzato@unipd.it (M.B.); antonio.paoli@unipd.it (A.P.); 2Department of Industrial Engineering, University of Padova, 35131 Padua, Italy; giuseppe.zullo@phd.unipd.it

**Keywords:** dynamic balance, postural control, center of pressure, movable plate, external perturbations, dynamometric platform

## Abstract

The translation of the base of support represents a promising approach for the objective assessment of dynamic balance control. Therefore, this study aimed to present a servo-controlled, electrically driven movable plate and a new set of parameters based on the center-of-pressure (CoP) trajectory. Twenty subjects were assessed on a force platform screwed over a movable plate that could combine the following settings: direction (forward (FW) and backward (BW)), displacement (25 mm, 50 mm, and 100 mm), and ramp rate (100 mm/s and 200 mm/s). The subjects underwent two sets of 12 trials randomly combining the plate settings. From the CoP trajectory of the 2.5 s time window after the perturbation, the 95% confidence-interval ellipse (Area95) and the CoP mean velocity (Unit Path) were calculated. Within the same time window, the first peak (FP), the maximal oscillations (ΔCoPMax), and the standard deviation (PPV) of the CoP anterior–posterior trajectory were calculated. The plate direction (*p* < 0.01), ramp rate (*p* < 0.001), and displacement (*p* < 0.01) affected the Area95, FP, and ΔCoPMax, while the Unit Path and PPV were influenced only by the ramp rate (*p* < 0.001) and displacement (*p* < 0.001). The servo-controlled, electrically driven movable plate and the CoP-related parameters presented in this study represent a new promising objective tool for dynamic balance assessment.

## 1. Introduction

Balance in humans involves a set of postural-control mechanisms [1] that jointly work to control body segments against gravity and to maintain the center of pressure (CoP) within the base of support [2]. Poor postural balance control is one of the major risk factors for falling in daily living activities [3], above all among the older adult population [4,5,6]. In a sports context, balance is one of the main determinants of performance [7], and no sport technical gesture can be efficiently achieved without optimal postural balance control [8,9]. Postural balance is governed by two control processes: the automatic process, in which the person is unaware of the adjustments of postural muscle tone, and the cognitive processes, in which self-body information and object localization in extra-personal space are needed [1]. The maintenance of the upright posture in an unperturbed environment (i.e., static balance) is ruled by the automatic processes at the brainstem and spinal cord level. In contrast, reactions to internal or external disturbances (i.e., dynamic balance) in oriented or non-oriented tasks are governed by cognitive processes and require motor outputs from the cerebral cortex to the brainstem and spinal cord [1]. Therefore, since static and dynamic postural balance control are ruled by the same structures but with different contributions, a complete postural balance assessment requires both static and dynamic tests [10,11].

Static postural balance while standing is mainly assessed through dynamometric platforms, which allow the calculation of CoP displacements from the forces recorded. This static test is well standardized in terms of the procedure [12], time duration and sample frequency [13], and CoP-related parameters [14]. Conversely, dynamic postural balance assessment includes many functional and quantitative tests. Functional tests (e.g., the Berg Balance Scale and the Timed Up-and-Go) are mostly used in clinical settings where subjects are scored on a test-specific scale [11,15]. An alternative method to objectively assess dynamic balance during functional tests is the use of accelerometers [16,17]. Conversely, quantitative tests provide external perturbations of several types imposed through different motion constraints in which subjects are scored by kinetic and kinematic parameters, the muscle excitation or quantification of the plate oscillations [11]. One of the most common experimental approaches is the disruption of stable equilibrium and the recording of consequent postural reactions [18]. Over the recent years, balance following perturbed events has been explored in several fields of application, such as under muscular fatigue [19], to quantify concussion status [20], and to evaluate the influence of a lower-limb exoskeleton [21]. The translation of the base of support represented a promising approach [18,22,23,24], but the methods were not comparable in terms of plate settings, and each study considered different postural-control parameters. Therefore, given the importance of evaluating dynamic balance control with objective and repeatable methods, the first goal of this study was to present a new movable plate that could be adjustable concerning displacement, velocity, and direction. The second goal was to calculate, through a force platform applied over the translational plate, a new set of CoP-related parameters to objectively quantify dynamic balance control consequent to base of support translations.

## 2. Methods

### 2.1. Subjects

The a priori power analysis calculation (G*Power 3.1.9.2, Heinrich Heine University, Düsseldorf, Germany) showed that a sample size of 17 participants and a medium effect size of 0.25 would provide a statistical power of 0.8. Therefore, for the present study, 21 subjects with no history of (i) orthopedic injuries in the last year, (ii) neurological diseases, (iii) vestibular pathologies, and (iv) non-corrected sight or hearing disorders were enrolled. One withdrew from the study the week before the testing session because of an ankle injury. Thus, 20 subjects (M = 11, F = 9; mean ± SD: 23.6 ± 1.9 years; 62.5 ± 9.5 kg; 1.70 ± 0.10 m) regularly completed the research protocol. The subjects were instructed on the experimental procedures before giving their written informed consent to participation. The study was performed in accordance with the Declaration of Helsinki and approved by the ethical committee of the Department of Biomedical Sciences, University of Padova.

### 2.2. Experimental Design

A servo-controlled, electrically driven movable system was developed for the tests in collaboration with the company EnginLAB srl (Padova, Italy). The system consists of an electro-actuated cylinder (EnginLAB srl, Padova, Italy) and a linear potentiometer (Penny & Giles, Dorset, UK) connected to a 135 cm × 135 cm plate, whose motion is allowed by two ball-type linear guideways (Figure 1A). A controller (RTC-9000, EnginLAB srl, Padova, Italy) allows the operator to set the desired displacement (mm) and ramp rate (mm/s) based on the performance of the electro-actuated cylinder and the linear potentiometer (Figure 1B). Specifically, the maximum payload, acceleration, velocity, and displacement of the system are 100 kg, 1 g, 800 mm/s, and 140 mm, respectively. A force platform (AMTI BP400600, Watertown, MA, USA) was screwed over the movable plate to derive the subjects’ CoP from the ground reaction forces (GRFs). The sampling frequency of the force platform was set to 200 Hz.

During the dynamic balance test, the subjects stood on the force platform with extended legs and arms placed naturally along their sides (Figure 1C). The feet width was equal to the distance between the two acromions. As with the standard static balance test procedures [12], the subjects were instructed to gaze vertically at a thin green line placed in front of them on a white wall at 80 cm to keep balance. We used an experimental setup in which the feet were flat on the ground and the knees were extended, limiting our study to postural movement without taking a step.

We outlined a cross-sectional design to test the effect of unexpected external perturbations on dynamic balance control. Specifically, the independent variables were (i) the direction of the plate shifting (i.e., forward (FW) and backward (BW)) with respect to the standing position of the subject; (ii) the plate displacements (i.e., 25 mm, 50 mm, and 100 mm); (iii) the plate ramp rate (i.e., 100 mm/s and 200 mm/s). The displacement and ramp rate ranges were based on a preliminary test involving five young subjects in which the administration of increasingly difficult perturbations allowed the identification of the highest displacement and the highest ramp rate (i.e., 100 mm at 200 mm/s) before a step was taken to regain balance. Overall, the subjects performed 15 randomly administered trials. These consisted of 12 plate-shifting trials combining different directions, displacements, and ramp rates and 3 no-perturbation trials. The no-perturbation trials were administered to prevent subjects from always expecting a perturbation. After a 5 min rest, each subject performed the 15 trials again. A random-number-generator software (https://it.piliapp.com/random/number/, accessed on 19 May 2023) defined the order of the trials in both testing sessions. Each trial lasted 60 s, and the plate shifting randomly occurred between the twentieth and fortieth seconds so that subjects could not predict when the perturbation would occur. The force platform recorded the GRF signal for the whole trial duration.

### 2.3. Data Analysis

Given the directions of the plate perturbations (i.e., FW and BW), we defined the perturbation point (PP) as the instant the plate started to move. The analysis of the CoP trajectory focused on the calculation of the following parameters over a 2.5 s time window immediately after the PP (Figure 2A,C): Area95 (the area of the 95th percentile ellipse measured in cm^2^) and Unit Path (the path length per time unit, i.e., the average CoP velocity measured in cm∙s^−1^) [14]. Then, we calculated three additional parameters to deepen the postural responses along the direction of the perturbation (i.e., anterior–posterior) (Figure 2B,D): the first peak (FP), the maximal oscillations (ΔCoPMax), and the post-perturbation variability (PPV). The FP represents the difference between the maximal peak reached by the CoP displacement after the external perturbation and the mean value of the anterior–posterior CoP displacement before the PP. The ΔCoPMax is the sum of the first body reaction (FP) and its counterbalance (second peak, SP) consequent to the external perturbation. To quantify the variability of the anterior–posterior CoP displacement in the early phase after the external perturbation occurred, we calculated the standard deviation of the CoP trajectory over the 2.5 s time window from the PP.

### 2.4. Statistical Analysis

The data were averaged between the two trials for each experimental condition and analyzed with the statistic package software JASP 0.16.4.0 (University of Amsterdam, Amsterdam, The Netherlands). A three-way, repeated-measure analysis of variance (ANOVA) was run to investigate the significant main effects of the plate direction (i.e., FW and BW), plate ramp rate (i.e., 100 mm/s and 200 mm/s), and plate displacement (i.e., 25 mm, 50 mm, and 100 mm) on the CoP-related dynamic parameters. The assumption of sphericity was checked through Mauchly’s sphericity test, and when it was violated, the Greenhouse–Geisser correction was applied. The partial eta squared (η^2^_p_) was calculated to measure the effect size of the single variables in the ANOVA model. The post hoc comparisons were corrected using the Bonferroni method. The level of significance was set to *p* < 0.05.

## 3. Results

The Area95 (Figure 3A) was influenced by the plate direction (F = 8.123; *p* < 0.01; η^2^_p_ = 0.299), ramp rate (F = 112.365; *p* < 0.001; η^2^_p_ = 0.855), and displacement (F = 20.209; *p* < 0.001; η^2^_p_ = 0.515). Post hoc comparisons among the displacements highlighted significant differences between 25 mm and 100 mm (*p* < 0.001) and between 50 mm and 100 mm (*p* < 0.001). In addition, the ANOVA showed significant interactions between the displacement and the ramp rate (F = 10.822; *p* < 0.001; η^2^_p_ = 0.363) and between the ramp rate and the direction (F = 7.243; *p* < 0.01; η^2^_p_ = 0.276).

Considering the Unit Path (Figure 3B), the main effects of the plate ramp rate (F = 182.187; *p* < 0.001; η^2^_p_ = 0.906) and the displacement (F = 127.949; *p* < 0.001; η^2^_p_ = 0.871) were observed. Post hoc comparisons among the displacements showed statistically significant differences (*p* < 0.001) between 25 mm and 50 mm, 50 mm and 100 mm, and 25 mm and 100 mm. In addition, the ANOVA identified a significant interaction between the displacement and the ramp rate (F = 36.810; *p* < 0.001; η^2^_p_ = 0.660).

For the FP (Figure 4A), the results showed significant main effects of the plate direction (F = 47.657; *p* < 0.001; η^2^_p_ = 0.715), ramp rate (F = 178.513; *p* < 0.001; η^2^_p_ = 0.904), and displacement (F = 15.802; *p* < 0.001; η^2^_p_ = 0.454). Post hoc comparisons among the displacements showed a statistically significant difference between 25 mm and 50 mm (*p* < 0.001) and between 25 mm and 100 mm (*p* < 0.001). In addition, the ANOVA identified significant interactions between the displacement and the ramp rate (F = 28.215; *p* < 0.001; η^2^_p_ = 0.598) and between the ramp rate and the direction (F = 45.523; *p* < 0.001; η^2^_p_ = 0.706).

The ΔCoPMax (Figure 4B) was affected by the plate direction (F = 13.571; *p* < 0.01; η^2^_p_ = 0.417), ramp rate (F = 32.502; *p* < 0.001; η^2^_p_ = 0.631), and displacement (F = 25.151; *p* < 0.01; η^2^_p_ = 0.221). Post hoc comparisons among the displacements showed a statistically significant difference between 25 mm and 50 mm (*p* < 0.01) and between 25 mm and 100 mm (*p* < 0.05). In addition, the ANOVA identified significant interactions between the displacement and the ramp rate (F = 5.265; *p* < 0.05; η^2^_p_ = 0.217) and between the ramp rate and the direction (F = 32.943; *p* < 0.001; η^2^_p_ = 0.634).

The results of the PPV (Figure 4C) highlighted the significant main effects of the ramp rate (F = 26.080; *p* < 0.001; η^2^_p_ = 0.579) and the displacement (F = 40.022; *p* < 0.001; η^2^_p_ = 0.678). Post hoc comparisons among the displacements showed a statistically significant difference (*p* < 0.001) between 25 mm and 50 mm, 25 mm and 100 mm, and 50 mm and 100 mm. In addition, the ANOVA identified significant interactions between the displacement and the ramp rate (F = 12.820; *p* < 0.05; η^2^_p_ = 0.403) and between the ramp rate and the direction (F = 37.462; *p* < 0.001; η^2^_p_ = 0.663).

Table 1 and Table 2 summarize for all the pairwise comparisons the effect size of the differences (Cohen’s d) and 95% confidence intervals (95% CIs).

## 4. Discussion

In the present study, we presented a new electronic movable plate able to simulate slips as well as sudden accelerations or decelerations of a moving surface (e.g., tripping on a bus or slipping on a carpet) to study dynamic balance control. The system was designed by analyzing similar devices already developed in antecedent research [22,23,24,25,26], and the main goal was to enclose all the major characteristics of previous systems in a single instrumentation. Specifically, we adopted an electro-actuated cylinder able to move the plate at a higher displacement and ramp rate than the existing systems [22,23,24]. Then, the controller adopted allowed setting of the perturbation displacement to one of two waveforms (i.e., ramp and sine waveform), since muscle onset latency is affected by different displacement waveforms [26]. Setting all the descriptors of the perturbation (i.e., ramp rate, displacement, and displacement waveform) is fundamental to permit the accurate replication of results and to ensure the appropriate interpretation of research findings when comparing results generated from different laboratories [26]. 

The second goal of the study arose from the observation that dynamic balance assessment includes a broad number of tests, most of them characterized by non-standard methods and subjective scoring [11]. Moreover, objectively recording body reactions following external perturbations has long been considered a recognized experimental method for understanding motor-control mechanisms [18]. Thus, we introduced a new set of CoP-related parameters to score the performance of dynamic balance control, sensitive to different plate directions (i.e., FW and BW), ramp rates (i.e., 100 mm/s and 200 mm/s), and displacements (i.e., 25 mm, 50 mm, and 100 mm). The first two parameters presented are the Area95 and the Unit Path that consider both the anterior–posterior and the medio-lateral CoP displacements. They are calculated in a 2.5 s window immediately after the perturbation point. Similar to the static balance assessment [14], the Area95 was assumed to be representative of the overall dynamic postural performance (i.e., the smaller the ellipse area, the better the balance performance), while the Unit Path enclosed the efficiency of the postural-control system (i.e., the slower the velocity, the more efficient the dynamic postural control). The second set of CoP-related parameters (i.e., FP, ΔCoPMax, and PPV) refers to the CoP anterior–posterior component, and their definition took into account the model of neural pathways involved in the cortical control of automatic postural responses to external perturbations [27,28]. 

Given that postural responses last hundreds of milliseconds, the authors hypothesized that brainstem circuits initiate a response that is subsequently modified by cortical circuits during its later phases [27]. Therefore, the FP and ΔCoPMax values mostly reflect the efficacy of the earliest feet-in-place postural responses of stabilizing muscles consequent to the external perturbation (i.e., the lower the values, the better the postural responses). Conversely, the PPV calculated over the 2.5 s window includes the voluntary postural strategies to reduce the anterior–posterior CoP oscillations after the external perturbation occurred. In other words, the lower the PPV, the faster the subject can regain a pre-perturbation balance condition. Besides the integrity of the structures of the central nervous system, effective body reactions to external perturbations are related to the contribution, and thus the efficiency, of the sensory system (e.g., visual, vestibular, and proprioception systems). Given these premises, the new set of CoP-related parameters showed good sensitivity to all the tested conditions. In detail, when the ramp rate and displacement increased, the attendant worsening of the Area95, Unit Path, FP, ΔCoPMax, and PPV parameters highlighted a decrease in dynamic balance performance. 

Similarly, Runge and colleagues [22] showed that the whole-body center-of-mass position obtained from kinematic analysis was displaced in magnitude as the translation velocity increased in backward ramp displacement. More recently, Zemkovà and colleagues [24] reported that the center-of-mass velocity was more influenced by plate velocity than by plate displacement. Separate consideration should be made of the effect of the plate direction (i.e., FW and BW) on the magnitude of FP, ΔCoPMax, and PPV. Indeed, all the parameters were lower in the FW than in the BW condition. Our results partially agree with those of a previous study in which the same trend was observed but only at the highest plate velocity [24]. In this regard, the human bipedal quiet stance has been modeled as a single inverted pendulum whose pivot is located at the ankle. In this model, the projection of the center of mass on the ground falls in front of the ankle (i.e., 50–60 mm), creating a dorsiflexor moment around the ankle [29]. It follows that along the anterior–posterior axis, the CoP distance from the anterior margin of the base of support (i.e., the tiptoes) is bigger than the CoP distance from the posterior margin (i.e., the heels). This condition allows the CoP to move more anteriorly than posteriorly without exceeding the limit of the base of support.

In conclusion, the characteristics of the electro-actuated cylinder and the CoP-related parameters derived from the force platform, make the whole system a promising tool for the neurophysiological study of human postural responses [28] and balance rehabilitation [30]. Moreover, considering that approximately 50% of falls among older adults are thought to be due to the sudden motion of the base of support [31], these systems might find application in fall risk assessment among frail populations [32].

The present study has some potential limitations to acknowledge. First, although the servo-controlled, electrically driven movable system presented in this study allows the implementation of several sets of external perturbations and waveforms, it is limited to a maximal ramp rate and displacement of 800 mm/s and 140 mm, respectively. Next, the set of new CoP-related parameters objectively quantifies the balance performance, but it is not suitable to discriminate between the main postural strategies (i.e., ankle or hip strategy) adopted by the subjects. Moreover, the present study investigated only linear measures of the CoP time series; future research might also incorporate non-linear measures to include the dynamic characteristics of the stabilogram in balance assessment.

## 5. Conclusions

In conclusion, the new servo-controlled, electrically driven movable system and the set of CoP-related parameters presented in this study represent a new promising objective tool for dynamic balance assessment. The areas of employment of the tool could be multiple, including, among many others, the study of aging-induced postural disorders or of neurological pathologies on postural control, the assessment of specific training programs for fall risk prevention, and the effect of sensory contribution on dynamic balance control. Moreover, it can find applications in cross-sectional studies to compare different groups of subjects (e.g., athletes and sedentary individuals) and in longitudinal studies to assess the efficacy of balance-oriented training or rehabilitation protocols.

## Figures and Tables

**Figure 1 sensors-23-06203-f001:**
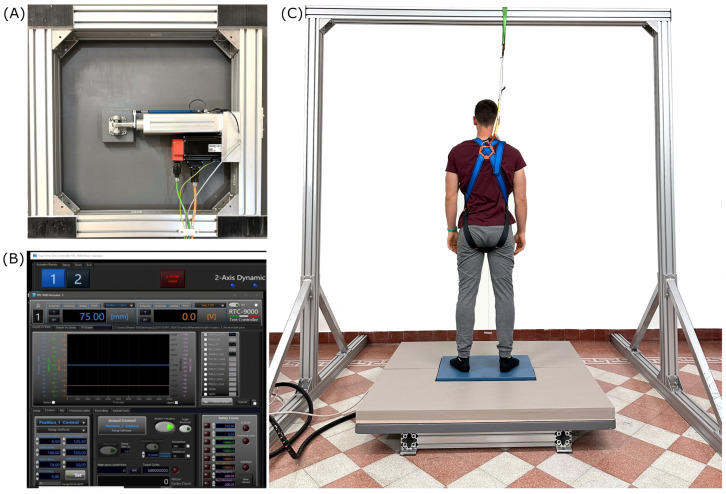
Experimental setting: (**A**) bottom view of the electro-actuated cylinder and the linear potentiometer used to move the 135 cm × 135 cm plate over the aluminum frame with ball-type linear guideways; (**B**) software interface of the RTC-9000 controller adopted to set the plate parameters; (**C**) overall view of the experimental setting.

**Figure 2 sensors-23-06203-f002:**
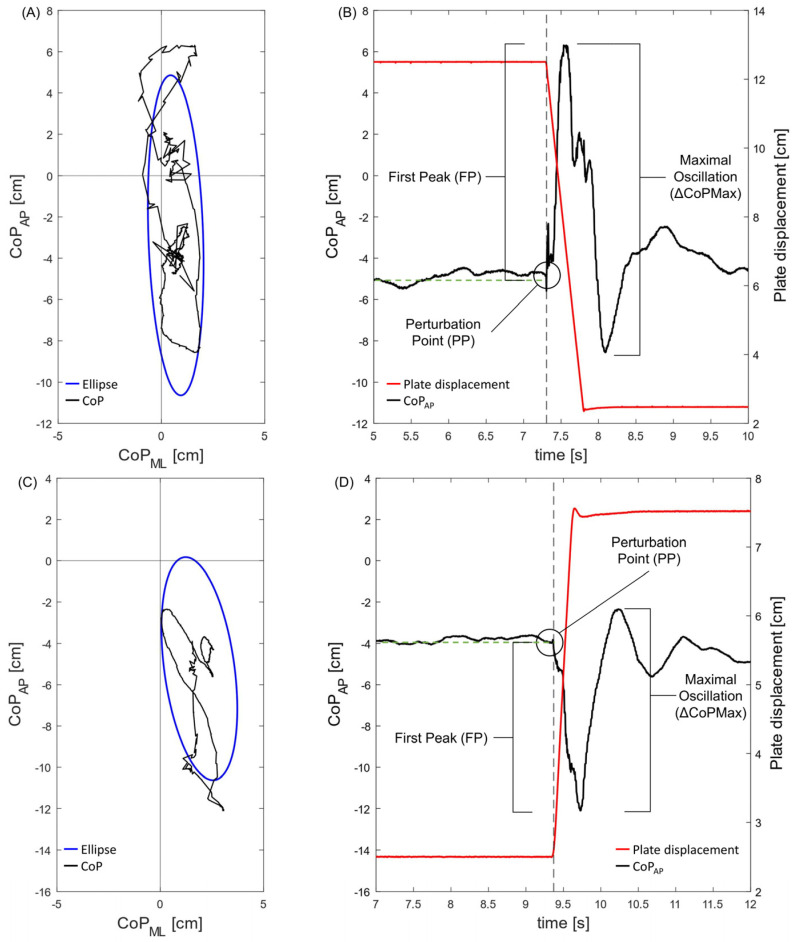
Graphical examples of the CoP-related parameters in backward ((**A**,**B**) displacement: 100 mm; ramp rate: 100 mm/s) and forward ((**C**,**D**) displacement: 50 mm; ramp rate: 200 mm/s) plate shifting. (**A**,**C**) CoP trajectory within the 2.5 s time window (in black) and 95th percentile ellipse (in blue). (**B**,**D**) First peak (FP) and maximal oscillation (ΔCoPMax) refer to the anterior–posterior CoP trajectory; the gray dotted line marks the perturbation point (PP), and the green dotted line marks the mean value of the CoP trajectory before the PP (see text for more details).

**Figure 3 sensors-23-06203-f003:**
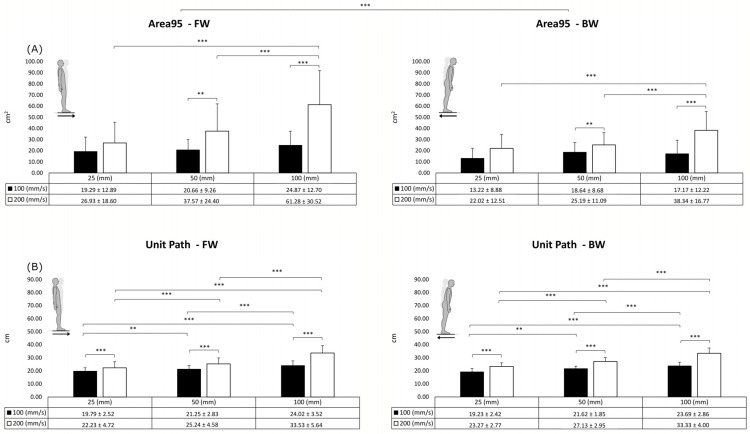
Results of Area95 (**A**) and Unit Path (**B**) in the forward (FW) and backward (BW) directions. Data are presented as mean ± standard deviation. ** (*p* < 0.01); *** (*p* < 0.001).

**Figure 4 sensors-23-06203-f004:**
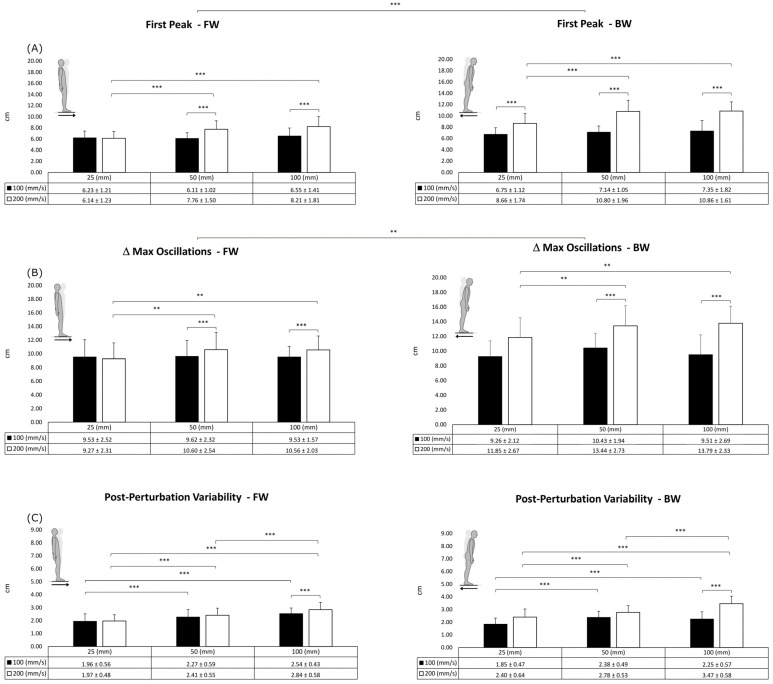
Results of the first peak (**A**), maximal oscillations (**B**), and post-perturbation variability (**C**) parameters in the forward (FW) and backward (BW) directions. Data are presented as mean ± standard deviation. ** (*p* < 0.01); *** (*p* < 0.001).

**Table 1 sensors-23-06203-t001:** Effect size of the differences (Cohen’s d) and 95% confidence intervals (95% CIs) calculated for the Area95 and Unit Path in pairwise comparisons.

	Area95		Unit Path	
	Cohen’s d	95% CI	Cohen’s d	95% CI
**Direction**				
FW vs. BW	−0.63	[−16.19, −2.48]	−0.16	[−0.66, 1.40]
**Ramp rate (mm/s)**				
100 vs. 200	−2.37	[−19.45, −13.03]	−3.01	[−6.76, −4.94]
**Displacement (mm)**				
25 vs. 50	−0.47	[−11.17, 0.87]	−1.25	[−3.87, −1.48]
25 vs. 100	−1.39	[−21.07, −9.02]	−3.52	[−8.70, −6.32]
50 vs. 100	−0.92	[−15.92, −3.87]	−2.27	[−6.02, −3.64]

**Table 2 sensors-23-06203-t002:** Effect size of the differences (Cohen’s d) and 95% confidence intervals (95% CIs) calculated for the first peak (FP), the maximal oscillations (ΔCoPMax), and the post-perturbation variability (PPV) in pairwise comparisons.

	FP		ΔCoPMax		PPV	
	Cohen’s d	95% CI	Cohen’s d	95% CI	Cohen’s d	95% CI
**Direction**						
FW vs. BW	1.54	[1.22, 2.29]	0.82	[0.66, 2.39]	0.39	[−0.03, 0.41]
**Ramp rate (mm/s)**						
100 vs. 200	−2.98	[−2.37, −1.73]	−1.27	[−2.65, −1.22]	−1.14	[−0.62, −0.25]
**Displacement (mm)**						
25 vs. 50	−0.93	[−1.61, −0.40]	−0.68	[−1.90, −0.19]	−1.12	[−0.61, −0.20]
25 vs. 100	−1.19	[−1.90, −0.69]	−0.57	[−1.72, −0.01]	−1.99	[−0.93, −0.52]
50 vs. 100	−0.26	[−0.89, 0.31]	0.11	[−0.68, 1.30]	−0.86	[−052, −0.11]

## Data Availability

Data sharing is not applicable to this article.

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
