# Peer review of "Center of Pressure Behavior in Response to Unexpected Base of Support Shifting: A New Objective Tool for Dynamic Balance Assessment"

_sensors, 2023, doi:10.3390/s23136203_

Round 1
Reviewer 1 Report
Dear authors,
Thank you for your work on developing the platform motion device and CoP evaluation method for balance assessment.
This report focuses on developing a device for evaluating dynamic balance and processing CoP data to assess automatic postural responses to balance loss.
While the manuscript is well-written, it lacks a thorough analysis comparing the findings with previous research. Additionally, the processing method of the presented CoP data should be clearly defined for better understanding.
So, the manuscript can be improved in the following aspects:
Q1) The authors introduced a paper on dynamic balance control using platform movement in Lines 58-59. However, since the introduced papers are at least 7 and up to 26 years old, it is necessary to introduce and compare the latest papers on dynamic balance control. For example, we consider the following papers to be references:
- Ilaria, M., et al. "Feasibility and application of the B.E.A.T. tested for asessing the effects of lower limb exoskeletons on human balance." (2022), https://doi.org/10.3390/robotics11060151
- Carla, D. P. R., et al. "Instantaneous interjoint rescaling and adaptation to balance perturbation under muscular fatigue." (2019), https://doi.org/10.1111/ejn.14606
- Deborah, J., et al. "Towards defining biomarkers to evaluate concussions using virtual reality and a moving platform (BioVRSea)." (2022), https://doi.org/10.1038/s41598-022-12822-0
- Taylor, W. C., et al. "Postural threat modulates perceptions of balance-related movement during support surface rotations." (2019), https://doi.org/10.1016/j.neuroscience.2019.02.011)
Q2) Lines 81-82 mention a cylinder to move the plate and a linear potentiometer. The equipment's model name and manufacturer (city and country) must be specified.
Q3) For more precise analysis, the direction of the plate shifting, plate displacement, and plate ramp rate should be specified in the description of the CoP path analysis graph in Figure 2. In addition, if the CoP graph in Figure 2 is a forward-direction disturbance, the difference between the two disturbances can be clearly understood if the CoP graph for the backward-direction disturbance is additionally disclosed and an explanation is added.
Q4) According to Figure 2, Area 95, defined by the author, does not contain the movement of COP_AP. A mathematical method for measuring Area 95 must be described in the method. Also, we need to discuss CoP trajectories that fall outside the ellipse or fall short of the ellipse range, as shown in Figure 2 (A).
Q5) The figure's description should be placed at the bottom of the image.
Reviewer 2 Report
Dear Authors:
First of all, I would like to congratulate you for the research developed. It is an interesting study, of clinical and scientific impact and well developed. However, the manuscript has certain formal and scientific limitations that need to be addressed before possible publication:
ABSTRACT: Correct, nothing to contribute.
INTRODUCTION: An excessively succinct version of this section has been provided. In addition, there is a lack of further development of the different methods of postural control assessment, developing their different benefits and drawbacks (doi: 10.3390/su12031222 // 10.1080/21679169.2017.1347707).
METHODS: The methodology employed is adequate and is described in detail. However, the manuscript would increase its scientific quality if it complemented the data provided with the calculation of the sample size and effect size of the sample analyzed.
RESULTS: This section should be complemented with graphical elements to facilitate the interpretation of the figures provided.
DISCUSSION: This section also suffers from a lack of bibliographic references with which to contrast and support the arguments provided.
CONCLUSIONS: The use of bibliographic references in this section is totally inadequate and discouraged.
Kind regards
Reviewer 3 Report
comments made directly on the document (attached)

Reviewer 4 Report
Abstract
Line 15: don’t start with numerics.
Introduction
Lines 58-60: expand a bit more on the research gap.
Methods
Line 106: is there any reason for the 15 plus 3 trials?
Line 133-142:
1) do you think it is pertinent to test a sex effect?;
2) why partial eta and not eta squared for the effect size (see: Ferguson, C. J. (2009). An effect size primer: A guide for clinicians and researchers. Professional Psychology: Research and Practice, 40(5), 532–538);
3) what about the effect size for the pairwise comparison? Cohen’s d?
Results
Add the F-ratios over the results section.
In figures to what correspond the white and black columns? It´s hard to understand.
In the figures/tables referring to the pairwise comparison, add the mean difference, 95CI, and effect size
Discussion
Overall, it is well written. But, just as you did in the conclusion, I think that here in the discussion a paragraph expanding about practical applications would be helpful.
Minor editing
Round 2
Reviewer 1 Report
N/A
Reviewer 2 Report
Dear Authors,
After the work carried out to improve and correct the manuscript, I now consider that the study is suitable for publication in this Journal.
Kind regards
Reviewer 4 Report
The authors improved the manuscript which is now clearer.
Minor editing.